# SSR: Alignment-Aware Modality Connector for Speech Language Models

## Abstract

Fusing speech into pre-trained language model (SpeechLM) usually suffers from inefficient encoding of long-form speech and catastrophic forgetting of pre-trained text modality. We propose SSR-Connector (Segmented Speech Representation Connector) for better modality fusion. Leveraging speech-text alignments, our approach segments and compresses speech features to match the granularity of text embeddings. Additionally, we introduce a two-stage training pipeline that includes the distillation and fine-tuning phases to mitigate catastrophic forgetting. SSR-Connector outperforms existing mechanism for speech-text modality fusion, consistently achieving better speech understanding (e.g., +10 accuracy on StoryCloze and +20 on Speech-MMLU) while preserving pre-trained text ability.

## 1 Introduction

Large language models (Brown et al., 2020; Chowdhery et al., 2022; Chiang et al., 2023; Anil et al., 2023; Touvron et al., 2023; OpenAI et al., 2024, LLMs) have demonstrated remarkable performance across various tasks and extending pre-trained abilities from LLMs to other modalities has sparked interest in multimodal LLMs (Alayrac et al., 2022; Liu et al., 2023b; OpenAI et al., 2024; Tang et al., 2024; Défossez et al., 2024). In this work, we focus on integrating speech into pre-trained language models (SpeechLMs). A straightforward approach is to transcribe speech into text and use these transcriptions as prompts for large language models (Huang et al., 2023); however, such cascaded systems suffer from error propagation, higher latency, and cannot leverage raw speech information like emotion, speaker identity, and other paralinguistic cues (Faruqui & Hakkani-Tür, 2021; Lin et al., 2022; Kim et al., 2024). Consequently, developing end-to-end SpeechLMs that directly fuse speech or audio input has gained popularity, where various approaches have been explored to encode speech and align its representation with pre-trained language models (Zhang et al., 2023; Rubenstein et al., 2023; Yu et al., 2023; Maiti et al., 2024; Hassid et al., 2024a; Tang et al., 2024; Nguyen et al., 2024).

Speech representations can be integrated into pre-trained language models mainly through two approaches. The first method involves using connector modules that align speech representations with the language model's input space without modifying the model's existing vocabulary. These connector-based techniques typically incorporate a compression module to shorten the speech features, enhancing efficiency. However, connectors are generally first trained for the speech recognition task (with concatenated speech-to-text data) and **lack the ability to support text or speech generation unless further instruction-finetuned**. The second approach, unit-based fusion, directly incorporates discrete speech units—normally derived from self-supervised models like HuBERT (Hsu et al., 2021), XLS-R (Babu et al., 2021), or DinoSR (Liu et al., 2023a)—into the language model's vocabulary. This allows the language model to be fine-tuned with a combination of speech and text tokens, enabling it to handle dual-modal inputs and outputs. Despite its versatility, **unit-based fusion can lead to longer and less efficient training contexts** due to the sparser nature of speech information. Regardless of the fusion approach, SpeechLMs often face the challenge of catastrophic forgetting, where the model loses its pre-trained text capabilities (Tang et al., 2024; Nguyen et al., 2024; Défossez et al., 2024).

To tackle these challenges, we propose SSR-Connector (Segmented Speech Representation Connector), which grounds speech representations in the same semantic space as transcription token embeddings. Different from prior work that concatenates speech with text (Fig. 1 (a,b)) for modality fusion, we leverage speech-text alignments to segment and compress speech features (Fig. 1 (c)), resulting in representations that match the length of text tokens.

To mitigate catastrophic forgetting when introducing the speech modality, we propose a two-stage training pipeline. In Stage 1, we freeze the LLM and pre-train the connector using speech-text distillation, adapting speech inputs into compressed representations semantically aligned with text embeddings. In Stage 2, we unfreeze the LLM and fine-tune it using next-token prediction, with the adapted representation as input and the corresponding transcription tokens as targets.

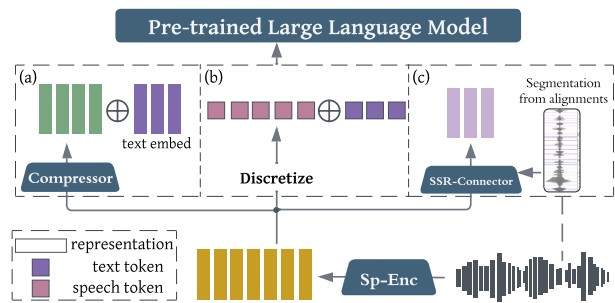

Figure 1: Comparison of different approaches for speech-text modality fusion. *(a)*: compressor-based connector. *(b)*: direct fusion with speech units. *(c)*: our alignment-aware connector.

SSR-CONNECTOR outperforms previous SpeechLMs (e.g., SPIRITLM (Nguyen et al., 2024), VOXTLM (Maiti et al., 2024), TWIST (Hassid et al., 2024b), AUDIOLM (Borsos et al., 2023)) on tasks including Prompt-based Automatic Speech Recognition (ASR), Spoken Language Understanding (sWUGGY (Nguyen et al., 2020), sBLIMP (Nguyen et al., 2020), and StoryCloze (Mostafazadeh et al., 2017)), Massive Multitask Language Understanding (Hendrycks et al., 2021, MMLU), and Speech-MMLU (our synthesized speech variant of MMLU to assess cross-modal understanding). Additionally, we provide detailed analyses of speech-text aligners (§4.3) and fine-tuning mechanisms (§5) to offer best practices when using SSR-CONNECTOR for modality fusion.

## 2 RELATED WORK

**Modality Fusion for Speech Language Models** SpeechLM typically encodes audio waveforms into high-dimensional features using pre-trained encoders and integrate these representations to pre-trained LLMs via a connection (adapter) module (Wu et al., 2023; Yu et al., 2023; Zhang et al., 2023; Tang et al., 2024). To compress speech representations, Fathullah et al. (2023) apply stacking-based fixed-rate compression on speech features extracted from the Conformer model (Gulati et al., 2020). Inspired by the Q-former architecture (Li et al., 2023a), Yu et al. (2023) compress speech features using a fixed number of query tokens, while Tang et al. (2024) extend this approach to a window-level Q-former to support variable frame-rate reduction. Alternatively, Wu et al. (2023) utilize Connectionist Temporal Classification (CTC) (Graves et al., 2006) to compress representations.

Besides connector-based modality fusion, pre-processing other modalities—such as speech, vision, and videos—into tokens (Lyu et al., 2023; Li et al., 2023b; Team, 2024; Kondratyuk et al., 2024) has attracted attention for its scalability. Speech units are typically extracted from self-supervised representations (Hsu et al., 2021; Babu et al., 2021; Chung et al., 2021; Liu et al., 2023a). For instance, AudioLM (Borsos et al., 2023) integrates semantic tokens from w2v-BERT (Chung et al., 2021) and acoustic tokens from SoundStream (Zeghidour et al., 2021), modeling them autoregressively for audio generation. Rubenstein et al. (2023) fine-tune the pre-trained LLM PaLM-2 (Anil et al., 2023) with audio tokens processed by AudioLM, enabling both text and speech as input and output. Similarly, VoxtLM (Maiti et al., 2024) performs multi-task training with speech units and text tokens, achieving high-quality speech recognition and synthesis. To mitigate catastrophic forgetting, Nguyen et al. (2024) propose an interleaved training mechanism to fuse speech tokens into LLAMA2 model.

**Speech-text Alignment Extraction** Various aligner tools are available for extracting speech-text alignments. For example, the Montreal Forced Aligner (MFA) (McAuliffe et al., 2017) is an easy-to-use tool based on the Kaldi toolkit (Povey et al., 2011). Connectionist Temporal Classification (CTC) (Graves et al., 2006) is also widely used for speech-text alignment (Sainath et al., 2020; Huang et al., 2024); since it is a by-product of speech recognition, it supports alignment without explicit text labels. More recently, the UnitY2 aligner (Communication et al., 2023) and the ZMM-TTS aligner (Gong et al., 2024) have shown excellent alignment performance across multiple languages. These aligners rely on speech units extracted from pre-trained encoders (Baevski et al., 2020; Hsu et al., 2021; Babu et al., 2021) and use variants of RAD-TTS (Shih et al., 2021) as their alignment backbone.

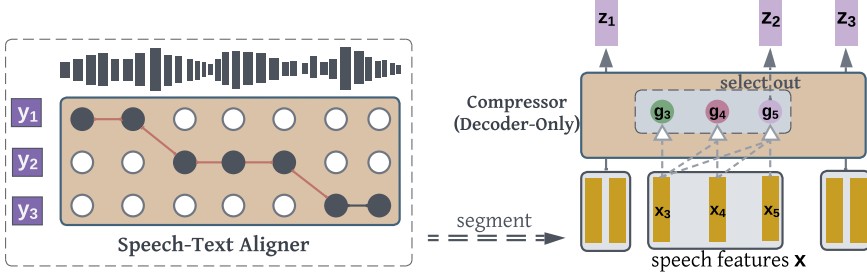

Figure 2: SSR-CONNECTOR compresses speech features using speech-text alignments. Features are transformed by a Decoder-only model and selected at boundary index of each segment.

## 3 METHODOLOGY

We develop an alignment-aware speech representation connector to foster modality fusion between speech and pre-trained language model. We introduce our connector in §3.1, with detailed descriptions of its aligners in §3.2. Lastly, we present the two-stage training pipeline for our connector in §3.3.

### 3.1 ALIGNMENT-AWARE SPEECH REPRESENTATION CONNECTOR

Though previous connectors (Fathullah et al., 2023; Yu et al., 2023; Wu et al., 2023; Tang et al., 2024) vary in their compressor designs, they do not explicitly leverage speech-text alignment information. SSR-CONNECTOR, in contrast, uses speech-text alignments to segment and compress speech features into the same granularity as text tokens. As illustrated in Fig. 2, our connector consists of two components: (1) a speech-text aligner and (2) a feature compressor.

Given speech features $x = (x_1, \cdots, x_n) \in \mathbb{R}^{n \times D}$ extracted by pre-trained speech encoders (e.g., WAV2VEC2.0, HUBERT, WHIPSER, etc.), the aligner produces a monotonic mapping (alignment path) between the speech features and their transcriptions $y = (y_1, \cdots, y_m) \in \mathbb{R}^{m \times 1}$. This mapping can be computed based on both speech features (or their units) and transcriptions (Communication et al., 2023; Gong et al., 2024), or solely based on speech input (Sainath et al., 2020; Dong & Xu, 2020; Huang et al., 2024) (see §3.2 for details). Using the alignment mapping, we segment the input into $m$ chunks of speech features, where each chunk semantically corresponds to a transcription token. For example, in Fig. 2, speech features are segmented at indices $(2, 5, 7)$ according to the alignment path. We refer to these indices as boundary indices.

Once the boundary indices are identified, we first apply a linear layer to transform the speech features to match the embedding dimension $H(H > D)$ of the pre-trained LLM, since LLMs typically have a larger feature dimension than pre-trained speech encoders. We then use the boundary indices to aggregate and compress the speech representations in each chunk through a Transformer Decoder model (Vaswani et al., 2017). Specifically, we apply a causal decoder-only model to transform the speech features into high-dimensional representations $g = f(x; \theta_{\text{dec}}) \in \mathbb{R}^{n \times H}$. Given that features at later positions include information from prior positions, we employ a selection-based compression method that takes the transformed features $g$ at the boundary indices to form the compressed representation $z \in \mathbb{R}^{m \times H}$. Although our initial design included a block-wise attention mask to restrict information flow within each chunk (as shown in Fig. 2, where the middle segment's features do not attend to previous segments), we found that removing these masks simplifies training and inference with minimal impact on performance, as detailed in §4.4.

### 3.2 SPEECH-TEXT ALIGNERS

We extract speech-text alignment with various aligners to segment speech features and we provide a brief overview of various aligners we experimented below:

**UnitY2 Aligner**  The UnitY2 aligner (Barrault et al., 2023) is a forced aligner that computes speech-text alignment using discrete speech units and character-level text tokens. The speech units are derived by applying K-Means clustering to the XLS-R model (Babu et al., 2021). The aligner is trained jointly with a non-autoregressive text-to-unit (T2U) model, adopting the architecture of the RAD-TTS model (Shih et al., 2021) but replacing the target mel-spectrogram with speech units. It first

computes a soft-alignment $A^{\text{soft}} \in \mathbb{R}^{V \times U}$ between the characters and units:

$$D_{i,j} = ||s_i^{\text{char}} - s_j^{\text{unit}}||_2, \tag{1}$$

$$A_{i,j}^{\text{soft}} = \frac{e^{-D_{i,j}}}{\sum_k e^{-D_{k,j}}} + P_{\text{prior}}(i|j), \tag{2}$$

where $s^{\text{char}}$ and $s^{\text{unit}}$ are the outputs of the character and unit encoders, respectively (both encoders consist of an embedding layer and a 1D convolution layer). $D \in \mathbb{R}^{V \times U}$ is a distance matrix with $V$ and $U$ representing the vocabulary sizes of characters and speech units. $P_{\text{prior}} \in \mathbb{R}^{V \times U}$ is the Beta-binomial alignment prior matrix to encourage near-diagonal paths (Shih et al., 2021). After soft-alignment is computed, the monotonic alignment search (MAS) algorithm Kim et al. (2020) is applied to extract the most probable monotonic alignment path.

**CTC-based Aligner**  Since the UnitY2 aligner requires both speech and transcription, it does not support streamable alignment extraction. To enable textless alignment computation, we explored two CTC-based (Graves et al., 2006) aligners. Given the speech features $x$ and text sequences $y$, CTC computes $P(y|x)$ by summing over all valid alignment paths:

$$P(y|x) = \sum_{\pi \in \mathcal{B}^{-1}(y)} P(\pi|x) \tag{3}$$

Here, $\pi$ denotes a possible alignment path that maps to the target sequence $y$, and $\mathcal{B}^{-1}(y)$ represents the set of all valid paths that collapse to $y$ after removing blanks and repeated labels. We investigated two CTC variants: one using character-level text sequences (CHAR-CTC) and another using subword token sequences (SUB-CTC), which shares the same vocabulary as the LLM model.

**CIF-based Speech Connector**  For both CTC and UnitY2 aligners, we extract segmentations from the alignments and then apply selection-based compression. We also experimented with Continuous Integrate-and-Fire (Dong & Xu, 2020, CIF) as the connector, which is designed to learn segmentation and perform compression simultaneously. Instead of relying on a fixed, pre-computed segmentation, CIF dynamically segments and aggregates speech features by scoring each feature and computing a weighted average. For more details, we refer readers to the original paper (Dong & Xu, 2020).

### 3.3 TRAINING METHOD

Previous approaches to integrate speech into LLMs typically use speech-text data concatenated in ASR format (i.e., speech representation followed by its transcription text embedding), to pre-train the connector (Yu et al., 2023; Wu et al., 2023; Tang et al., 2024). However, after such pre-training, the model is limited to speech recognition task and necessitates another instruction-tuning stage to perform generative tasks with pre-trained connectors (Zhang et al., 2023; Tang

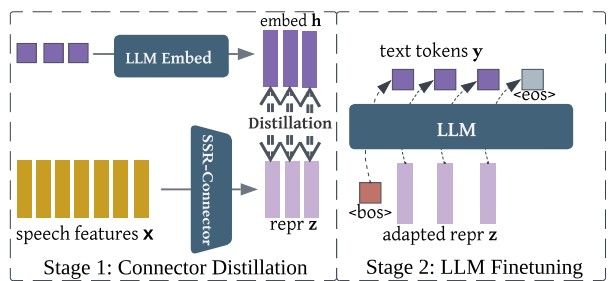

Figure 3: Two-stage training pipeline for SpeechLM with our alignment-aware modality connector.

et al., 2024). Moreover, once the LLM is unfrozen and fine-tuned (whether based on a pre-trained connector or direct fusion with speech units), it suffers from catastrophic forgetting, leading to degraded text capabilities (Nguyen et al., 2024; Tang et al., 2024).

With SSR-CONNECTOR, we convert speech into representations with the same granularity as their transcription tokens. This allows us to fine-tune the SpeechLM directly using the next-token prediction objective, **where the input is the compressed representation $z$ and the target is the transcription $y$**. This approach is possible because our feature $z$ and text token $y$ share the same length $m$. However, our preliminary studies showed that directly fine-tuning with the next-token prediction objective leads to catastrophic forgetting, undermining the pre-trained LLM's abilities. Therefore, we propose a two-stage training pipeline consisting of a distillation stage and a fine-tuning stage, as shown in Fig. 3.

In Stage 1, we pre-train SSR-CONNECTOR by distilling the LLM's text embeddings to align the connector's representations with the LLM's embedding space. Formally, given aligned speech-text data, we compute the text embeddings $\boldsymbol{h} = f(\boldsymbol{y}; \theta_{\text{emb}})$, where $\boldsymbol{y}$ is the transcription token sequence, $\theta_{\text{emb}}$ is the embedding table, and $f$ maps tokens $\boldsymbol{y}$ to their embeddings. Following our connector design in §3.1, we obtain the compressed speech representations $\boldsymbol{z}$. For distillation, we use a combination of cosine similarity loss $\mathcal{L}_{\text{cos}}$ and mean squared error (MSE) loss $\mathcal{L}_{\text{MSE}}$

$$\mathcal{L} = \lambda\mathcal{L}\text{cos} + \mathcal{L}_{\text{MSE}} = \frac{1}{m}\sum_{i=1}^{m}\left[\lambda\left(1 - \frac{\mathbf{z}_i^\top\mathbf{h}_i}{|\mathbf{z}_i|\cdot|\mathbf{h}_i|}\right) + |\mathbf{z}_i - \mathbf{h}_i|^2\right], \tag{4}$$

where $\lambda$ is a hyperparameter to balance the losses[1]. In Stage 2, we fine-tune the LLM with the pre-trained speech connector using the next-token prediction objective. We freeze the speech connector and update only the LLM's parameters using the negative log-likelihood (NLL) loss:

$$\mathcal{L}_{\text{NLL}} = -\sum_{t=1}^{m}\log p(y_t \mid \boldsymbol{z}_{<t}; \theta_{\text{LLM}}) \tag{5}$$

where $y_t$ is the $t^{\text{th}}$ token in the transcription sequence $\boldsymbol{y}$, $\boldsymbol{z}_{<t}$ denotes all preceding speech representations, and $\theta_{\text{LLM}}$ represents the LLM's parameters. Note that our NLL loss is computed using only the preceding speech representations $\boldsymbol{z}_{<t}$ (see Fig. 3), whereas previous methods (Wu et al., 2023; Tang et al., 2024) condition on both speech information and preceding text tokens $\boldsymbol{y}_{<t}$.

We offer detailed descriptions of different aligners and demonstrate the performance of SpeechLM after distillation training in §4. In §5, we present results after fine-tuning SpeechLM and compare various fine-tuning strategies to identify the method that minimizes catastrophic forgetting.

## 4 STAGE 1: ALIGNMENT-AWARE CONNECTOR DISTILLATION

### 4.1 DATASETS

For distillation training, we use the aligned speech-to-text dataset MLS (Pratap et al., 2020), specifically the English portion, which consists of about 50,000 hours of speech. To evaluate our SpeechLMs, we employ several datasets as shown in Table 1. To assess the model's spoken language understanding (SLU) capabilities, we follow Nguyen et al. (2024) and use sWUGGY, sBLIMP, and the StoryCloze dataset. sWUGGY and sBLIMP are detailed in (Nguyen et al., 2020). Briefly, sWUGGY evaluates whether a model can discriminate between real spoken words and non-words (e.g., "brick" vs. "blick"), while sBLIMP assesses if the model can distinguish between a grammatically correct spoken sentence and its ungrammatical variant (e.g., "cats are lazy" vs. "cats is lazy"). We evaluate our SpeechLMs on both text ($T$) and speech ($S$) versions of sWUGGY and sBLIMP. The StoryCloze dataset measures whether the model can identify the plausible ending between two sentences given the beginning of a short story, which typically requires high-level semantic understanding and common sense (Mostafazadeh et al., 2017). Besides spoken and text versions of StoryCloze, following Nguyen et al. (2024), we use a speech-text version ($S \rightarrow T$), where the beginning of the story is synthesized into speech and the two ending sentences are kept in text format. This version requires the model to have cross-modal understanding to infer the sensible story ending.

MMLU (Hendrycks et al., 2021) is widely used to assess LLMs' knowledge comprehension, understanding, and reasoning abilities, and we use it to measure the extent of forgetting during cross-modal fine-tuning. Since MMLU is a diverse and high-quality evaluation dataset for LLMs, we craft a variant, Speech-MMLU, to assess our SpeechLM's cross-modal understanding. Specifically, we utilized AUDIOBOX (Vyas et al., 2023), a high-quality text-to-speech synthesizer, to convert the question portion of each choice task into speech while keeping the multiple-choice answers in text format. We selected a subset of MMLU to construct our Speech-MMLU dataset, as some domains' questions are not suitable for synthesis (e.g., the algebra subset contains many mathematical notations that are not synthesized properly). sWUGGY, sBLIMP, StoryCloze, and Speech-MMLU are all categorized

---

[1]In practice, we set $\lambda = 5$ to balance the scales of the cosine similarity and MSE losses

| Eval Dataset | Type | Eval Metric | Eval Format |
|---|---|---|---|
| sWUGGY (Nguyen et al., 2020) | Choice Task | Accuracy | $S, T$ |
| sBLIMP (Nguyen et al., 2020) | Choice Task | Accuracy | $S, T$ |
| StoryCloze (Mostafazadeh et al., 2017) | Choice Task | Accuracy | $S, T, S \to T$ |
| MMLU (Hendrycks et al., 2021) | Choice Task | Accuracy | $T$ |
| Speech-MMLU (*Ours*) | Choice Task | Accuracy | $S \to T$ |
| LibriSpeech (Panayotov et al., 2015) | Generation Task | Word Error Rate | $S \to T$ |

Table 1: Evaluation Datasets and their types. For the evaluation format, $S$ is speech-only, $T$ is text-only, and $S \to T$ means the evaluation prompt consists of speech prefix and text continuation.

as "Choice Task", meaning several choices are presented to the SpeechLM (Speech-MMLU has four choices while the other task has only two choices). For each task, we compute accuracy using groundtruth choice and the highest likelihood choice predicted by the SpeechLM.

Lastly, we also evaluate our SpeechLM's ASR performance using the Librispeech clean/other datasets. We evaluate ASR in a prompt-based fashion with zero-shot and five-shot setting. More details about our evaluation (e.g., prompts for ASR, Speech-MMLU construction, etc.,) can be found in Appendix.

### 4.2 MODEL SETUP

We instantiate our LLM using the pre-trained LLAMA3 model (Touvron et al., 2023) and employ DinoSR (Liu et al., 2023a) as our pre-trained speech feature extractor. Our speech connector includes a linear layer that maps DinoSR's extracted representations ($D = 768$) to the LLM's embedding space dimension ($H = 4096$). We then utilize a 4-layer Transformer Decoder to transform and compress the speech representations based on alignments, as described in §3.1. The compressed representations $z$ and the embeddings of text tokens $h$ are used to compute the distillation loss for updating the connector's parameters. We train our connector for 400,000 steps with a learning rate of $1 \times 10^{-5}$, using dynamic batching with a maximum of 4,096 tokens per device. We employ distributed data parallelism (DDP) with 32 A100 GPUs.

To extract alignments, we experimented with different aligners listed in §3.2. For the UnitY aligner[2], we used it off-the-shelf to construct alignment indices. Since the UnitY2 aligner provides alignments based on character-level tokens, we merge the durations into subword level to ensure that the compressed representations and text embeddings have the same granularity. For CTC-based aligners, we trained them using a 4-layer Transformer Decoder followed by a linear projection. In the character-level variant (CHAR-CTC), we deduplicate the sequence to obtain character-level durations and then merge them into subword-level durations to segment the speech features, similar to the UnitY2 aligner. In the subword-level variant (SUB-CTC), we directly use CTC's blank token to segment the speech input.

### 4.3 ALIGNER PERFORMANCE COMPARISON

To compare the quality of different aligners, we trained several SSR-CONNECTOR based on different aligners via distillation. We evaluated the aligners using the Librispeech clean test set by computing the Cosine Similarity (**Cos(%)**) and Mean Squared Error (**MSE**) between the compressed representations and text embeddings. Additionally, we performed zero-shot and five-shot ASR with the learned connector. Note that we never explicitly trained the model for ASR tasks, and the base LLM remained frozen during Stage 1 training. Therefore, the model achieves

| Model Type | Cos(%)↑ | MSE↓ | WER (%) ↓ |
|---|---|---|---|
| UNITY2 | **96.8** | **0.018** | **5.6 / 4.0** |
| CHAR-CTC | 95.1 | 0.023 | 9.7 / 6.5 |
| SUB-CTC | 92.2 | 0.037 | 16.7 / 14.0 |
| CIF | 77.5 | 0.096 | 27.6 / 23.7 |

Table 2: Performance comparison (with Cosine Similarity, MSE, and 0/5-shot ASR WER) between different aligners used for Stage 1 training, evaluated on Librispeech clean test set.

low word error rates (**WER**) only when the distilled speech representations closely resemble the text embeddings. As shown in Table 2, the UNITY2 aligner brings the speech representations close to their corresponding text embeddings, achieving very low WER in both zero-shot and five-shot ASR

---

[2] Publicly available at `https://github.com/facebookresearch/seamless_communication/blob/main/docs/m4t/unity2_aligner_README.md`

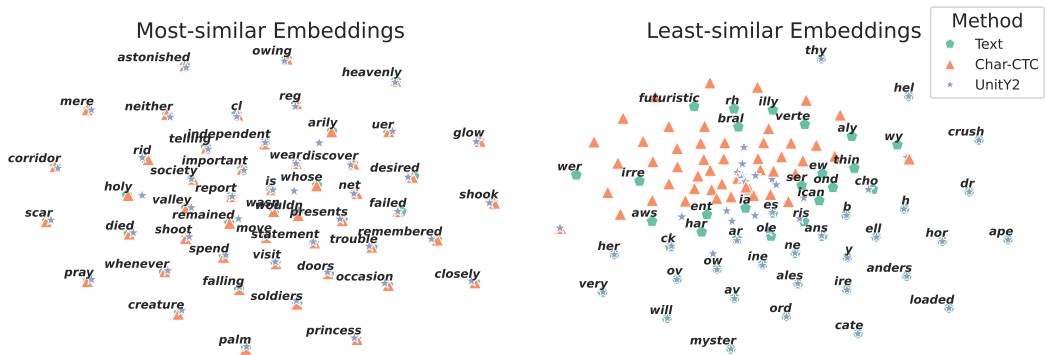

Figure 4: t-SNE plots of text and speech representations after distillation.

settings. Among textless aligners, we found that CHAR-CTC performs the best, likely because it has a much smaller vocabulary compared to SUB-CTC, making it easier to learn. Lastly, CIF resulted in suboptimal performance, possibly due to its less accurate alignment, as its segmentation is predicted by accumulating scores without exploiting the monotonicity between speech and text.

To visualize the effect of distillation, we present t-SNE plots of the adapted speech representations and text embeddings in Fig. 4, categorizing them into high and low similarity groups based on the cosine similarity between CHAR-CTC representations and text embeddings. We observe that longer subwords tend to exhibit higher similarity, likely because their long segments make it easier for the connector to convert speech representations into corresponding text embeddings. Furthermore, longer subwords possess more coherent semantics compared to shorter tokens like 'wy' or 'ia'.

Given that UNITY2 and CHAR-CTC performs the best, we also follow Huang et al. (2024) to measure their word boundary error (WBE) and word average duration (WDUR) using the TIMIT (Garofolo et al., 1993) data. Though the aligner quality can be further improved with other methods such as CTC + Label Prior (Huang et al., 2024), MMS (Pratap et al., 2023), or MFA (McAuliffe et al., 2017), CHAR-CTC and UNITY2 still achieve good quality and we choose them out of simplicity and general availability (unlike "CTC+Label Prior", for example, which requires customization with library like k2[3]).

| Aligner | WBE↓ | WDUR |
|---|---|---|
| Groundtruth | 0 | 305 |
| UNITY2 | 33 | 279 |
| CHAR-CTC | 42 | 230 |
| *Other Aligners* | | |
| CTC+Label Prior | 29 | 288 |
| MMS | 37 | 242 |
| MFA | 23 | 314 |

Table 3: Alignment quality of aligners.

### 4.4 EXPERIMENTAL RESULTS

In the previous section (§3.2), we compared different aligners and found that UNITY2 and CHAR-CTC performed the best. Consequently, we evaluate SpeechLM using these two aligners. First, we assess the model on Spoken Language Understanding (SLU) tasks and the MMLU benchmark (Hendrycks et al., 2021). We compare our model against several generative speech systems, all of which utilize Transformer-Decoder models trained on speech units. These methods vary in training approaches (pretrained from scratch or fine-tuned), types of speech units, and the size of training data.

Briefly, GSLM (Lakhotia et al., 2021) trains on speech units like HuBERT, TWIST (Hassid et al., 2024b) is a textually pretrained speech model based on Llama-13B (Touvron et al., 2023), and AudioLM (Borsos et al., 2023) employs a cascade system with a semantic sequence model alongside coarse- and fine-acoustic models. These models focus solely on speech without capabilities for text understanding or generation. More recently, SPIRITLM (Nguyen et al., 2024) and VoxtLM (Maiti et al., 2024) have adopted multi-task training objectives that incorporate text-only, speech-only, and speech-text token sequences to fuse the speech modality into pre-trained LLMs effectively. Since the original SPIRITLM is fine-tuned based on LLAMA2, we follow the same recipe to fine-tune the LLAMA3-based SPIRITLM ourself for a fair comparison on text relevant metrics like MMLU.

---

[3]https://github.com/k2-fsa/k2

| Model Type | sWUGGY | | sBLIMP | | Storycloze | | | MMLU |
|---|---|---|---|---|---|---|---|---|
| | T | S | T | S | T | S | S→T | 5-shot |
| *Previous Work* | | | | | | | | |
| GSLM$^\diamond$ (Lakhotia et al., 2021) | $\emptyset$ | 64.8 | $\emptyset$ | 54.2 | $\emptyset$ | 53.3 | $\emptyset$ | $\emptyset$ |
| AUDIOLM$^\diamond$ (Borsos et al., 2023) | $\emptyset$ | 71.5 | $\emptyset$ | 64.7 | $\emptyset$ | _ | $\emptyset$ | $\emptyset$ |
| VOXTLM$^\diamond$ (Maiti et al., 2024) | 80.3 | 66.1 | 74.2 | 57.1 | _ | _ | _ | _ |
| TWIST$^\diamond$ (Hassid et al., 2024b) | $\emptyset$ | **74.5** | $\emptyset$ | 59.2 | $\emptyset$ | 55.4 | $\emptyset$ | $\emptyset$ |
| MOSHI$^\clubsuit$ (Défossez et al., 2024) | $\emptyset$ | 72.6 | $\emptyset$ | 58.8 | $\emptyset$ | 60.8 | _ | 49.8 |
| SPIRITLM$^\diamond$ (Nguyen et al., 2024) | 80.3 | 69 | 73.3 | 58.3 | 79.4 | 61 | 64.6 | 36.9 |
| SPIRITLM (LLAMA3)$^\spadesuit$ | 77.6 | 73.5 | **74.5** | 56.3 | 75.1 | 61.1 | 61.6 | 53.5 |
| SSR-CONNECTOR | | | | | | | | |
| UNITY2 + Blockwise-mask | **81** | 71.5 | **74.5** | **73.1** | **80.9** | **71.8** | **75** | **65.3** |
| UNITY2 | 81 | 71.2 | 74.5 | 72.4 | 80.9 | 69.3 | 74.8 | 65.3 |
| CHAR-CTC | 81 | 56.4 | 74.5 | 67.3 | 80.9 | 62.2 | 74.3 | 65.3 |
| CHAR-CTC (Unit-based) | 81 | 54.1 | 74.5 | 61.8 | 80.9 | 59.2 | 72.5 | 65.3 |
| *Cascade System* | | | | | | | | |
| ASR (WHISPER) + LLAMA2 $^\diamond$ | 84.1 | 79.2 | 72.8 | 71.6 | 81.9 | 75.7 | 75.7 | 46.2 |

Table 4: Model performance on spoken language understanding and MMLU. $^\diamond$: Results taken from Nguyen et al. (2024).$^\clubsuit$: Results taken from Défossez et al. (2024). $^\spadesuit$: Our implementation of SPIRITLM based on LLAMA3 checkpoint. We fill with $\emptyset$ the task and modality that are not supported by the reported system, and with _ the scores that are not publicly available. We bold the best result and highlight the second-best system with the blue color box (excluding the cascaded system).

**Spoken Language Understanding Performance**    As shown in Table 4, our systems outperform previous models on all tasks except sWUGGY. The sWUGGY dataset includes incorrectly spoken words that cause segmentation errors because these words were not present during aligner training, leading to our system's lower performance on this dataset. However, sWUGGY is the least significant task since it relies on synthesized incorrect words and does not require the model's understanding or reasoning capabilities. In contrast, both UNITY2 and CHAR-CTC based connector greatly surpass previous models on other datasets, demonstrating the effectiveness of SSR-CONNECTOR in enhancing SLU performance while preserving model's text understanding ability.

Beyond UNITY2 and CHAR-CTC, we introduce two additional systems for ablation. The UNITY2 + Blockwise-mask system achieves the highest performance by applying a blockwise attention mask to further constrain the Transformer-Decoder (described in §3.1). However, due to its marginal improvement over UNITY2 and increased computational cost, we decide to simplify the design and remove the blockwise-attention masks. The CHAR-CTC (Unit-based) system differs by utilizing discrete speech units instead of raw waveform features processed by the DinoSR (Liu et al., 2023a) encoder. These units are extracted via K-Means clustering on DinoSR representations, which leads to some information loss during discretization and reconstruction, resulting in lower performance compared to CHAR-CTC. Nonetheless, CHAR-CTC (Unit-based) demonstrates that *our alignment-aware connector design is compatible with both continuous waveforms and discrete speech units*.

**Speech-MMLU and Prompt-based ASR Performance**    In addition to SLU tasks, we evaluate our systems on the Speech-MMLU benchmark, which assesses cross-modal understanding and is more challenging than previous SLU tasks. We also conduct prompt-based ASR evaluations to assess the quality of the adapted features. As shown in Table 5, our systems greatly outperform the previous SpeechLM (SPIRITLM), achieving a +20 accuracy improvement on the Speech-MMLU dataset[4]. These results indicate that SpeechLM based on SSR-CONNECTOR possesses enhanced cross-modal abilities that enable it to comprehend spoken questions and reason through multiple-choice options to select correct answers. Similarly, our systems achieve much lower WERs on both the Librispeech clean and other test sets compared to SPIRITLM. Notably, neither SPIRITLM nor our system were trained on ASR tasks, *so the model relies solely on in-context learning to generate transcriptions*. Even our weakest system (CHAR-CTC (Unit-based)) can outperform SPIRITLM 's 10-shot result.

---

[4] We report micro-average across 22 domains and the detailed breakdown is available in Appendix C.

| Model Type | Speech MMLU ↑ | | ASR Clean Test ↓ | | ASR Other Test ↓ | |
|---|---|---|---|---|---|---|
| | 0-shot | 5-shot | 0-shot | 5-shot | 0-shot | 5-shot |
| SPIRITLM (Nguyen et al., 2024) | N/A | N/A | N/A | 21.9* | N/A | 29.2* |
| SPIRITLM (LLAMA3) | 40.5 | 42.75 | N/A | 21.0* | N/A | 28.5* |
| SSR-CONNECTOR | | | | | | |
| UNITY2 + Blockwise-mask | **65.0** | **69.5** | **5.0** | **2.6** | **8.1** | **6.8** |
| UNITY2 | 64.2 | 68.6 | 5.6 | 4.0 | 12.1 | 10.6 |
| CHAR-CTC | 61.7 | 66.5 | 9.7 | 6.5 | 20.2 | 14.9 |
| CHAR-CTC (Unit-based) | 57.4 | 62.3 | 12.6 | 8.8 | 25.6 | 18.6 |

Table 5: Comparison of Speech-MMLU and ASR performance. Speech-MMLU results are micro-averages across all domains. *: For SPIRITLM, We report WER using 10-shot prompting, following Nguyen et al. (2024). N/A: We did not evaluate SPIRITLM in those settings.

## 5 STAGE 2: SPEECH LANGUAGE MODEL FINE-TUNING

In Stage 1 (§4), we freeze the pre-trained LLM and distill its text embeddings into our alignment-aware connector. In this section, we fine-tune SpeechLM by freezing the connector and updating the LLM. This process enhances the model's spoken language understanding (SLU) performance by fitting SpeechLM on the aligned speech-text data, albeit at the expense of degrading its pre-trained text capabilities. In the following sections, we compare various methods to mitigate catastrophic forgetting and demonstrate their trade-offs between speech and text understanding.

### 5.1 MITIGATE CATASTROPHIC FORGETTING

**Model and Dataset Setup**  We fine-tune SpeechLM using the next-token prediction objective described in §3.3. In this stage, we freeze the connector distilled in Stage 1 and unfreeze the LLM (LLAMA3) parameters. Following Stage 1 (§4), we use the MLS dataset for training and evaluate the model on the same speech and text understanding tasks. Beyond vanilla fine-tuning, we also explore Low-rank Adaptation (Hu et al., 2021, LoRA) and multitask fine-tuning as they have been shown effective for mitigating catastrophic forgetting in other tasks (Xue et al., 2021; Vu et al., 2022). Details of our fine-tuning setup are shown below:

- **Vanilla Fine-tuning**: We perform full fine-tuning on the aligned speech-text data with a learning rate of $1 \times 10^{-6}$ and a maximum token size of 4096. Training is model-parallelized across 32 A100 GPUs using Fully Sharded Data Parallel (Zhao et al., 2023, FSDP).
- **LoRA Fine-tuning**: We leverage the low-rank constraints from as regularization to prevent model overfitting in MLS dataset. We configure LoRA layers with $\alpha = 512$, $r = 256$, and a dropout probability of 0.1.
- **Multitask Fine-tuning**: To preserve the LLM's pre-trained text capabilities, we also fine-tune SpeechLM on text-only data using the standard Negative Log-Likelihood (NLL) loss. The dataloader is configured to sample from both speech-text and text-only datasets with equal probability. We continue using the MLS dataset for speech-text training and utilize a subset of the LLAMA2 training datasets (Touvron et al., 2023) for text-only training.

| Model Type | sWUGGY | | sBLIMP | | Storycloze | | | MMLU |
|---|---|---|---|---|---|---|---|---|
| | T | S | T | S | T | S | S→T | 5-shot |
| CHAR-CTC | 81 | 56.4 | 74.5 | 67.3 | 80.9 | 62.2 | **74.3** | **65.3** |
| + Vanilla Fine-tuning | 82.5 | 56.6 | 75.8 | 68.8 | 75.2 | 62.8 | 71 | 57.4 |
| + LoRA Fine-tuning | 82.4 | 56.5 | 75.8 | 68.7 | 76.3 | 62.6 | 71.5 | 58.2 |
| + Multitask Fine-tuning | **82.9** | **56.7** | **75.9** | **68.9** | **81** | **63.4** | 73.1 | 63.1 |

Table 6: Comparison of different Stage 2 fine-tuning methods (reported after fine-tuned for 5k updates). Multitask fine-tuning obtains the best improvement on SLU tasks while achieving least catastrophic forgetting. We bold the best performance and use blue color box for the second-best result.

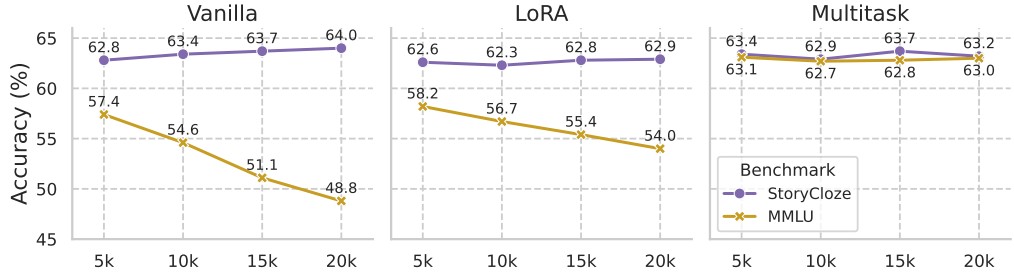

Figure 5: Comparison of different fine-tuning methods on StoryCloze ($S$) and MMLU benchmark.

| Model Type | Speech MMLU↑ | | ASR Clean Test↓ | | ASR Other Test↓ | |
|---|---|---|---|---|---|---|
| | 0-shot | 5-shot | 0-shot | 5-shot | 0-shot | 5-shot |
| SPIRITLM (LLAMA3) | 40.5 | 42.75 | N/A | 21.0* | N/A | 28.5* |
| CHAR-CTC | **61.7** | **66.5** | 9.7 | 6.5 | 20.2 | 14.9 |
| + Multitask Fine-tuning | 48.1 | 56.3 | N/A | **5.7** | N/A | **13.1** |

Table 7: Speech-MMLU and ASR performance of different models. *: For SPIRITLM, We report WER using 10-shot prompting for ASR, following Nguyen et al. (2024). N/A: The 0-shot generation of our fine-tuned SpeechLM tends to have hallucinations (keep generating after completing the transcription) so we only report its 5-shot performance.

## 5.2 COMPARISON OF FINE-TUNING METHODS

In Fig. 5, we compare different fine-tuning methods on StoryCloze ($S$) and MMLU. StoryCloze performance is indicative of how well model is fitted to the speech modality and MMLU measures the degree of catastrophic forgetting in pre-trained text abilities. We observe that Vanilla Fine-tuning quickly overfits to the speech domain, achieving improved performance on StoryCloze but drastically decreasing MMLU accuracy. In contrast, LoRA Fine-tuning introduces strong regularization, resulting in limited improvements in speech understanding. Although LoRA mitigates catastrophic forgetting to some extent compared to vanilla fine-tuning, performance still steadily declines. **Multitask fine-tuning emerges as the most promising approach**, enhancing speech understanding while largely mitigating catastrophic forgetting, evidenced by the modest 2-point drop in MMLU.

Since model performance does not further improve with additional training steps (as shown in Fig. 5), we utilize the checkpoint trained for 5,000 updates to compare with baseline models. The results are presented in Table 6 and Table 7. Note that even with only 5,000 updates, the model has observed all speech-text data due to our large effective batch size. Across SLU, MMLU, and ASR tasks, the fine-tuned SpeechLM outperforms baseline methods on tasks primarily relying on speech-only information (sWUGGY, sBLIMP, ASR), with multitask fine-tuning achieving the best performance among all fine-tuning methods. However, we also observe a decline in performance on $S \to T$ tasks such as Speech-MMLU and StoryCloze, indicating that **there is still unavoidable degradation of text capabilities** which adversely affects SpeechLM's cross-modal performance.

Overall, Stage 2 fine-tuning experiments highlight a trade-off between enhanced speech understanding and degraded text abilities when unfreezing pre-trained LLM weights. Though such forgetting phenomenon is unavoidable, our two-stage training pipeline has largely preserved SpeechLM's text ability and our experimental results underscore the importance of incorporating high-quality text data during cross-modal fine-tuning to balance performance across both modalities.

## 6 CONCLUSION

We propose SSR-CONNECTOR to inject speech representation into pre-trained LLMs. Through explicitly leveraging speech-text alignment, our connector compresses long and sparse speech information to the same granularity as text tokens. To mitigate catastrophic forgetting, we propose a two-stage training pipeline for modality fusion. Compared to previous baselines, our SpeechLM achieves much better speech understanding ability while retaining its pre-trained text ability.

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

# SUPPLEMENTARY MATERIAL

## A  DATASETS

As described in §4.1, we employ sWUGGY, sBLIMP, StoryCloze, MMLU, Speech-MMLU and Librispeech datasets to assess model performance. In this section, we provide more examples for each evaluation set. sWUGGY and sBLIMP are simple tasks where two choices can be directly compared. As shown in Table 8, sWUGGY provides two choices that requires models to discriminate real words from non-words. sBLIMP assesses whether model can distinguish between a grammatically correct sentence and its ungramatical variant. MMLU and StoryCloze, on the other hand, have a prefix and choices. The StoryCloze dataset measures whether the model can identify the logical ending between two sentences given the beginning of a short story. Since StoryCloze has a shared prefix, we can synthesize only the prefix part into speech and keep choices in text format, resulting in our $S \rightarrow T$ format evaluation that assess model's cross-modal understanding. Similarly, for MMLU, we also synthesize its prefix (the question portion) into speech and keep the choices in text format, resulting in our Speech-MMLU dataset. Since some topics have bad audio synthesis quality (e.g., the algebra subset contains many mathematical notations), we only keep 22 topics in our test suite (Table 9).

| Name | Prefix | Choices |
|------|--------|---------|
| sWUGGY | N/A | {Good=obsolete, Bad=odsolete} |
| sBLIMP | N/A | {Good=Walter was harming himself, Bad=Walter was harming itself} |
| StoryCloze | I had been giving this homeless man change every day. He was on the same corner near my house. One day, as I was driving through my neighborhood I saw a new car. Soon enough, I saw the same homeless man emerge from it! | {Good=I never gave the man money again. Bad=The next day I gave the man twenty dollars.} |
| MMLU | During the period when life is believed to have begun, the atmosphere on primitive Earth contained abundant amounts of all the following gases except | {"A": "oxygen", "B": "hydrogen", "C": "ammonia", "D": "methane"} |

Table 8: Examples of different evaluation datasets.

## B   EVALUATION METRIC AND PROMPT

Choice tasks (sWUGGY, sBLIMP, StoryCloze, MMLU, Speech-MMLU) are evaluated by comparing perplexity of different choices. The choice with smallest perplexity is selected as the prediction and we measure accuracy across different benchmarks.

For generation task (prompt-based ASR), we use the prompt below, with pairs of speech and transcription is provided to the SpeechLM. For 0-shot evaluation, we do not include any examplers.

> **Prompt**
>
> Given the speech, provide its transcription.
> [speech]: {demo speech}
> [text]: {demo transcription}
> ...
> [speech]: {speech to transcribe}
> [text]:

## C   SPEECH MMLU EVALUATION

We present the detailed comparison results in Table 9 for better comparison of model performance across different domains / topics. We see that the trend for different domains are mostly consistent, with our alignment-aware connector based on UNITY2 achieving the best performance, followed by CHAR-CTC based connector. Similar as our main findings, the unit-based system has worse performance due to information loss from discretization and the fine-tuned model suffers from catastrophic forgetting (albeit mitigated through our multitask fine-tuning approach). Nevertheless, all these SSR-CONNECTOR based system obtains better performance compared to SPIRITLM (LLAMA3), confirming the effectiveness of our modality-fusion strategy.

| Topic | SPIRITLM | | UNITY2 + Mask | | UNITY2 | | CHAR-CTC | | Unit-based | | Fine-tuned | |
|---|---|---|---|---|---|---|---|---|---|---|---|---|
| | 0-shot | 5-shot | 0-shot | 5-shot | 0-shot | 5-shot | 0-shot | 5-shot | 0-shot | 5-shot | 0-shot | 5-shot |
| Astronomy | 45.6 | 40.8 | 60.0 | 66.0 | 60.7 | 65.3 | 57.0 | 60.4 | 49.7 | 61.1 | 50.7 | 52.0 |
| Business Ethics | 37.1 | 40.2 | 52.0 | 60.0 | 53.0 | 62.0 | 56.0 | 59.0 | 52.0 | 55.0 | 37.0 | 51.0 |
| Clinical Knowledge | 36.0 | 39.8 | 60.6 | 63.3 | 61.0 | 62.9 | 61.2 | 62.7 | 57.8 | 57.4 | 47.3 | 53.8 |
| College Biology | 36.4 | 33.6 | 65.0 | 69.9 | 62.9 | 68.5 | 57.7 | 59.9 | 54.2 | 57.7 | 40.6 | 44.1 |
| Electrical Engineering | 37.7 | 44.2 | 52.5 | 57.4 | 52.5 | 53.9 | 48.2 | 58.9 | 44.7 | 48.2 | 53.2 | 54.6 |
| High School Biology | 40.8 | 41.2 | 66.0 | 72.2 | 67.6 | 72.2 | 63.3 | 68.2 | 57.1 | 65.6 | 50.5 | 62.5 |
| High School Gov. Pol. | 44.4 | 43.4 | 79.2 | 84.9 | 78.1 | 83.3 | 76.6 | 81.8 | 71.4 | 73.4 | 54.7 | 64.1 |
| International Law | 55.9 | 58.5 | 71.1 | 81.0 | 71.1 | 81.0 | 71.1 | 80.2 | 71.1 | 75.2 | 66.1 | 71.1 |
| Jurisprudence | 37.1 | 36.2 | 60.2 | 68.5 | 62.0 | 70.4 | 57.4 | 63.9 | 54.6 | 60.2 | 51.9 | 57.4 |
| Machine Learning | 39.3 | 32.1 | 45.8 | 59.3 | 50.8 | 59.3 | 45.8 | 61.0 | 44.1 | 57.6 | 39.0 | 55.9 |
| Management | 43.0 | 42.0 | 79.6 | 84.5 | 77.7 | 75.7 | 73.8 | 74.8 | 68.0 | 70.9 | 45.6 | 65.0 |
| Marketing | 39.8 | 49.8 | 77.8 | 85.0 | 76.1 | 81.6 | 76.9 | 81.6 | 74.4 | 76.9 | 51.3 | 67.1 |
| Miscellaneous | 38.5 | 36.4 | 69.2 | 71.5 | 66.6 | 70.1 | 60.3 | 64.6 | 52.3 | 57.5 | 42.7 | 50.3 |
| Moral Disputes | 39.1 | 42.3 | 59.5 | 66.5 | 59.5 | 67.3 | 56.4 | 62.7 | 52.9 | 62.1 | 43.6 | 52.9 |
| Nutrition | 45.0 | 47.3 | 68.4 | 69.1 | 66.1 | 66.8 | 65.5 | 62.8 | 64.5 | 59.8 | 52.8 | 58.5 |
| Philosophy | 37.5 | 37.2 | 58.3 | 64.5 | 59.0 | 62.5 | 55.9 | 64.1 | 54.6 | 59.5 | 44.0 | 53.1 |
| Prehistory | 38.9 | 43.3 | 62.0 | 66.4 | 61.1 | 64.5 | 61.2 | 64.3 | 55.0 | 57.5 | 49.1 | 55.2 |
| Security Studies | 43.8 | 54.8 | 63.8 | 67.8 | 61.7 | 67.8 | 68.1 | 76.9 | 59.3 | 69.2 | 51.0 | 59.7 |
| Sociology | 37.4 | 45.5 | 71.6 | 74.6 | 68.7 | 74.6 | 69.7 | 73.6 | 68.2 | 72.1 | 57.7 | 66.2 |
| US Foreign Policy | 56.7 | 60.8 | 80.0 | 80.0 | 78.0 | 85.0 | 75.8 | 81.8 | 75.8 | 83.8 | 61.0 | 76.0 |
| Virology | 40.1 | 46.3 | 47.9 | 49.1 | 49.1 | 53.9 | 47.9 | 49.7 | 46.1 | 51.5 | 46.7 | 44.8 |
| World Religions | 39.3 | 46.4 | 66.1 | 67.8 | 63.2 | 63.7 | 52.0 | 59.1 | 51.5 | 60.8 | 40.9 | 50.3 |
| Micro Average | 40.5 | 42.7 | 65.0 | 69.5 | 64.2 | 68.6 | 61.7 | 66.5 | 58.1 | 63.3 | 49.0 | 57.5 |

Table 9: Detailed Speech-MMLU evaluation results on different domains.

