# OpenReview forum: "SSR: Alignment-Aware Modality Connector for Speech Language Models"
_ICLR.cc/2025/Conference — Submitted to ICLR 2025_

### Official Review · Reviewer_j1u2 · 2024-10-29

**Soundness:** 3
**Presentation:** 3
**Contribution:** 2
**Rating:** 6
**Confidence:** 4

**Summary:**

This paper presents a method to improve speech langauge models by using speech-text alignment. Experiments have been conducted to evaluate the performance of the proposed method.

**Strengths:**

1. The idea of utilizing the alignement between speech and text sequences to build speech LMs is novel to me.
2. The paper is written clearly and easy to follow.

**Weaknesses:**

Since the speech-text alignement is considered in the proposed method, it has similarities with the ASR+LLM (cascade) implementation of speech LMs. For an ASR model, its task is to derive the aligned text from given speech. In Table 4, the performance of the cascade system was quite good. So, some discussions on comparing the proposed method with cascade systems should be emphasized.

**Questions:**

1) In Section 3.1, what is the unit of the transcription sequence y for alignement? characters, phonemes or words?
2) In Eq. (4), why are both similarity loss and MSE loss used? How to determine the weight for combining them?
3) In Section 5, why is the Char-CTC model used for evaulation since it didn't perform as good as UnityY2 in Section 4?
4) In Table 4, LLaMa2 was used for the cascade system while LLaMa3 was used in the proposed method. This seems unfair.

---

> ### Author Response · Authors · 2024-11-19
> **Response by Authors**
>
> We deeply appreciate the time and effort you have dedicated to reviewing this paper. We value your insights, and in response to your points of concern, we offer the following explanations:
>
>
> > …some discussions on comparing the proposed method with cascade systems should be emphasized.
>
> Thank you for raising this concern. We will add further motivation to clarify how our system differs from the cascaded system. The primary advantage of an end-to-end system is that the language model is exposed to encoded speech information, rather than just textual information. This allows an end-to-end system to leverage both semantic and paralinguistic information. We kindly refer you to our "general response to all reviewers" for additional experiments we conducted on the non-semantic aspects of our model.
> In this work, we focus on the semantic aspect of SpeechLM because there are still gaps between end-to-end systems and cascaded systems in terms of speech and text understanding. Our goal is to bridge these gaps with our modality fusion approach.
>
>
> > what is the unit of the transcription sequence y for alignement?
>
> The unit of transcription sequence is subword level (from Llama3’s tokenizer). We kindly refer you to line 298-305 for details.
>
> > why are both similarity loss and MSE loss used? How to determine the weight for combining them?
>
> The use of Cosine and MSE loss is an empirical choice. During training, we observed that the MSE loss initially decreases to about 0.1, after which the Cosine loss begins to decrease. We found that using Cosine loss in conjunction with MSE loss provides better guidance. This is because Cosine loss encourages speech representations to maintain similar relational structures to text embeddings, rather than focusing solely on the absolute differences in embedding vectors, as optimized by MSE. To determine the appropriate weight, we monitored the scale of both MSE and Cosine loss after some training and selected $\lambda =5$ to ensure that the Cosine loss is on the same scale as the MSE loss.
>
> > Why is the Char-CTC model used for evaulation since it didn't perform as good as UnityY2 in Section 4?
>
> The findings of our stage fine-tuning should generalize to different aligners. We choose Char-CTC because it might be more useful in practice as it does not require transcripts to compute the alignment.
>
>
> > LLaMa2 was used for the cascade system while LLaMa3 was used in the proposed method. This seems unfair.
>
> We acknowledge that using the Llama3 based cascaded system is more fair. However the difference between Llama2 and Llama3 on the speech understanding benchmarks (sWUGGY, sBLIMP, and Storycloze) will be marginal because these benchmarks’ text sentences are not hard for text-only models. Therefore, we simply re-use the results from the SpiritLM paper to save computational efforts as the primary comparison is made between our SSR-based system and prior SpeechLM.

---

> ### Author Response · Authors · 2024-11-28
> **Official Comment by Authors**
>
> Dear reviewer j1u2,
>
> We sincerely appreciate your valuable suggestions for improving our paper and hope our response has effectively addressed your concerns. Should you have any additional suggestions or questions, please do not hesitate to let us know. We would greatly appreciate it if you could consider raising the rating. Thank you!
>
> Best regards, Authors

---

### Official Review · Reviewer_Bk9J · 2024-11-01

**Soundness:** 2
**Presentation:** 3
**Contribution:** 2
**Rating:** 5
**Confidence:** 4

**Summary:**

The paper presents SSR-Connector, an innovative approach to integrating speech data into pre-trained LLMs by aligning and compressing speech representations to match text embeddings, enhancing speech-text modality fusion. This technique tackles two main challenges: inefficient processing of long speech sequences and the risk of catastrophic forgetting of text-based capabilities. SSR-Connector employs a two-stage training process: an initial distillation phase aligns speech representations with text embeddings, followed by fine-tuning to improve cross-modal understanding. The approach incorporates alignment-aware segmentation of speech features, optimized to work with text-compatible granularity, and achieves some gains in some speech understanding benchmarks, surpassing prior frameworks such as SpiritLM and VoxtLM.

**Strengths:**

1. The paper is well-structured and systematically describes each component of the SSR-Connector. Additionally, it clearly explains the methodology used to prevent catastrophic forgetting and offers comparisons with prior models.
2. The SSR-Connector introduces a novel "alignment-aware" approach to speech-text modality fusion.
3. The paper provides a comprehensive analysis of the proposed SSR-Connector across a variety of benchmarks, including StoryCloze, sWUGGY, and Speech-MMLU.

**Weaknesses:**

1. One significant aspect of SpeechLMs is their ability to capture information in speech beyond just the content. This has been analyzed in many related studies; for instance, in SALMONN [1] and Qwen2-Audio [2], paralinguistic information, such as emotion, is treated as an important evaluation aspect, and SpiritLM also examines capabilities related to understanding paralinguistic information. In addition, as the authors mention in the introduction, cascade systems are unable to learn paralinguistic information in speech, which is an advantage of end-to-end SpeechLMs. However, this paper lacks an analysis of the proposed model’s understanding of paralinguistic information and focuses solely on performance from a content comprehension perspective. If the model lacks the ability to understand paralinguistic information, then even if it addresses the issues of long speech sequences and catastrophic forgetting, it still wouldn't make for an ideal SpeechLM.

2. From the perspective of model methodology, it is challenging for the approach to capture paralinguistic information from speech. The methodology focuses on understanding speech content while overlooking other information within the speech. For instance, in stage 1, the target is text embedding, and achieving the best result would mean that the model could directly predict text embedding from speech but ignore other information. In stage 2, the SSR-Connector is fixed, and only the LLM is trained, so the LLM is also unable to learn information beyond content. Therefore, the entire training process essentially resembles a speech recognition model’s functionality by learning to extract textual information from speech. This limits the novelty of the paper, as the method does not introduce a more comprehensive SpeechLM capable of learning beyond content and instead functions more like an end-to-end model which integrates a speech recognition model with an LLM.

3. Another concern might be that the model’s best performance relies on the UNITY2 model for alignment, which itself requires both speech and transcription for alignment results. Given that a transcript is already available, why not use it directly as the LLM’s input instead of processing it through the SSR-Connector? In my view, using a model that doesn’t rely on transcription for alignment would better align with the scenario proposed in the paper.

4. As far as I know, the test sets used in the paper, such as sWUGGY and sBLIMP, are those employed in the Zero Resource Speech Challenge, where one of the key functions of the models is speech continuation. All the baseline models compared in the paper include speech generation capabilities. I would suggest comparing more models that, like the proposed model, take speech as input and output text, such as SALMONN [1], Qwen-Audio [3], and Qwen2-Audio [2].

[1] Tang, Changli, et al. "Salmonn: Towards generic hearing abilities for large language models." arXiv preprint arXiv:2310.13289 (2023).

[2] Chu, Yunfei, et al. "Qwen2-audio technical report." arXiv preprint arXiv:2407.10759 (2024).

[3] Chu, Yunfei, et al. "Qwen-audio: Advancing universal audio understanding via unified large-scale audio-language models." arXiv preprint arXiv:2311.07919 (2023).

**Questions:**

I have outlined my main concerns in the Weaknesses part. Here are the corresponding questions:
1. How does the proposed model account for paralinguistic information (e.g., emotions) in speech, and why has the analysis of this aspect been omitted from the paper, despite its significance in related studies?
2. Given the methodology’s focus on content comprehension, how does the model address the capture of paralinguistic information?
3. Why does the model rely on the UNITY2 alignment method, which uses both speech and transcription, rather than directly using the available transcription as input for the LLM? Would a model that does not depend on transcription for alignment be more aligned with the paper's goals?
4. Could you include more baseline models, such as SALMONN and Qwen2-Audio?

---

> ### Author Response · Authors · 2024-11-19
> **Response by Authors**
>
> We deeply appreciate the time and effort you have dedicated to reviewing this paper. We value your insights, and in response to your points of concern, we offer the following explanations:
>
> > However, this paper lacks an analysis of the proposed model’s understanding of paralinguistic information and focuses solely on performance from a content comprehension perspective.
>
> Thanks for raising the concern over paralinguistic information and we have conducted some additional analysis to show that, even though our focus is on semantics for modality fusion, the paralinguistic information can be preserved through speech encoder and utilized by the SpeechLM. We kindly refer you to our “general response to all reviewers” for more details.
>
> However, given that many prior SpeechLMs still lacks behind cascaded system in terms of speech/text understanding, we believe that better modality fusion approach that improves speech understanding and mitigates catastrophic forgetting over text modality is meaningful, even if only semantic (content comprehension) is considered.
>
>
> >  It is challenging for the approach to capture paralinguistic information from speech.
>
> We respectfully disagree with the assertion regarding the model’s ability to capture paralinguistic information. Firstly, since speech representation is integrated into language models, paralinguistic information is retained, as demonstrated by our additional analysis using the Expresso dataset. The key challenge lies in teaching models to utilize this information effectively and improving the encoding of paralinguistic information within the speech encoder. Our analysis shows that the model can capture some paralinguistic information through in-context learning, without requiring emotion-relevant fine-tuning or expressive tokens.
>
> Secondly, considering how other works, such as Salmonn, enable emotion recognition, their connector pre-training step also focuses primarily on semantics, using ASR format data to pre-train the Q-former based connector. They then teach models to utilize paralinguistic information through instruction fine-tuning. Our proposal is a foundational SpeechLM that emphasizes content comprehension, and our method is complementary to any instruction fine-tuning (Salmonn, Qwen2-audio) or special expressive token approach (SpiritLM) aimed at utilizing paralinguistic information (e.g., for emotion recognition or expressive speech generation).
>
> > In my view, using a model that doesn’t rely on transcription for alignment would better align with the scenario proposed in the paper.
>
> We agree with your point that an aligner is more practical if it does not rely on transcription, which is why we develop CTC and CIF-based aligners for the SSR-connector. We kindly refer you to Section 3.2 (lines 174-177), where we discuss our motivation for using a CTC-based aligner to support textless alignment computation. In all of our experiments, we discussed the CTC-based SpeechLM and demonstrated its superior performance over prior systems in both spoken language understanding and text understanding.
>
> > Could you include more baseline models, such as SALMONN and Qwen2-Audio?
>
> Thanks for the suggestion and we acknowledge that SALMONN and Qwen2-Audio are indeed capable speech-text systems. To alleviate your concern, we benchmark SALMONN over Storycloze (S->T) and Speech MMLU and compare it with our system. We kindly refer you to our “general response to reviewers” for details for results.
>
> However, we would like to emphasize that our focus in the paper is on how to improve the connector-based modality fusion for better foundational SpeechLM, instead of how to better instruction-finetune a SpeechLM for various downstream tasks as focused by SALMOON or Qwen2-Audio. Therefore, it is more reasonable for us to compare with prior systems on speech/text understanding benchmarks.

---

> ### Comment · Reviewer_Bk9J · 2024-11-25
>
> Thank you very much to the author for the efforts made in the rebuttal. However, I still have the following questions:
>
> > However, given that many prior SpeechLMs still lacks behind cascaded system in terms of speech/text understanding, we believe that better modality fusion approach that improves speech understanding and mitigates catastrophic forgetting over text modality is meaningful, even if only semantic (content comprehension) is considered.
>
> I completely agree with the author that catastrophic forgetting and modality fusion are critical topics in SpeechLM. However, the proposed approach focuses solely on content and heavily relies on ASR data throughout the training process, which essentially rules out the possibility of using other tasks for training. Therefore, it’s not simply a matter of "even if only semantic (content comprehension) is considered," but rather that evaluation is restricted to the semantic level alone.
>
> > Firstly, since speech representation is integrated into language models, paralinguistic information is retained, as demonstrated by our additional analysis using the Expresso dataset.
>
> Thank you to the author for conducting the additional experiment. Even if the author demonstrates in the experiment that SSR-Connector can capture emotion information (although I believe it is not as effective as Qwen-Audio and SALMONN in this aspect), its capability is still very limited and does not prove that SSR-Connector performs well in understanding elements beyond content. Additionally, other paralinguistic information, such as age and gender, is likely even more challenging to capture effectively.
>
> > Our proposal is a foundational SpeechLM that emphasizes content comprehension, and our method is complementary to any instruction fine-tuning (Salmonn, Qwen2-audio) or special expressive token approach (SpiritLM) aimed at utilizing paralinguistic information (e.g., for emotion recognition or expressive speech generation).
>
> First, the foundational SpeechLM is not solely focused on content comprehension. The first phase of training for both Qwen-Audio and Qwen2-Audio includes various speech-related tasks. Second, your argument is not very convincing because it does not demonstrate whether this is indeed a good foundation model. Specifically, the model after the instruction fine-tuning phase is more important and practically valuable, but you have not provided experimental evidence that SSR-Connector can retain its speech/text understanding capabilities after instruction fine-tuning. Finally, you claim that your proposal is a foundational SpeechLM, but I cannot find the paper provides any explanation to state this point, which contradicts your response.

---

> ### Author Response · Authors · 2024-11-25
> **Response by Authors**
>
> We appreciate your additional responses and questions. Before addressing them, we'd like to invite you to review our work from a different mindset.
> It appears that you have a preconceived notion of an ideal system (a capable SpeechLM that can perform multiple tasks, understand both semantic and paralinguistic information) based on prior work such as Salmonn or Qwen-audio. You then compare our work to this ideal system, highlighting aspects where we fall short (e.g., emotion recognition, multitask pre-training). However, our focus is entirely different. Since our submission is not a product/system demo, we propose a more suitable way to assess our paper:
> - Review our problem definition: Is the speech-text modality fusion problem worth investigating?
> - Review our method (SSR-connector): Does it effectively address the identified problem?
> - Review our experiments/analysis: Do our empirical results demonstrate the efficacy of the proposed method, and do our analyses provide new insights into the problem?
>
> To challenge our method, we suggest focusing on aspects where our approach may not adequately address the raised issue (modality fusion, catastrophic forgetting, etc.). Avoid introducing new problems (e.g., emotion recognition or multitask pre-training) that are not within our scope to solve. If you wish to hold us to a different standard by redefining/expanding the problem scope to paralinguistic information or instruction-finetuning stage, you must first discredit the importance of the problem we are solving (i.e., show that modality fusion or catastrophic forgetting is a trivial or already-solved problem for the scientific community).
>
> Now we provide detailed response to your questions below:
>
> > However, the proposed approach focuses solely on content and heavily relies on ASR data throughout the training process, which essentially rules out the possibility of using other tasks for training.
>
> Since ASR data has the largest quantity (due to the ease to synthesize them), it is natural to focus on this data format for pretraining (and previous capable systems like Moshi, Salmonn, SpiritLM, TWIST, etc., all focus on ASR data for pretraining). Additionally, our method works with any kind of speech-text, speech-only, or text-only input data, so please provide clarification on the “other tasks” you are referring to here.
>
> > its capability (on non-semantic aspect) is still very limited…other paralinguistic information, such as age and gender, is likely even more challenging to capture effectively.
>
> we provide our few-shot experiments on Expresso to show that non-semantic information is retained and we also stress that our method is complementary to approaches that facilitate model understanding beyond semantics. Since the non-semantic aspect is not the focus of our study, it is beyond the scope for us to prove our method can outperform previous studies in this aspect.
>
>
> > ...your argument is not very convincing because it does not demonstrate whether this is indeed a good foundation model.
>
> Please refer to our evaluation on spoken language understanding and MMLU.
>
>
> > Specifically, the model after the instruction fine-tuning phase is more important and practically valuable...
>
> The notion of “instruction fine-tuning phase is more important and practically valuable” is very problematic (a pre-trained LLM like Llama3 is not useful compared to llama3-instruct, does this mean pre-training method and details of Llama3 are not worth publishing if no instruction tuning result is reported?).
>
> Please be aware that this is a paper submission, not launching a new product/speech-text system. Our paper aims to address and provide more insights on speech-text modality fusion in the context of a connector-based SpeechLM and we focus on the pre-training stage. Instruction-finetuning phase experiments are nice to have but should not be used as critism to reject the paper, similar to emotion/paralinguistic information.
>
>
> > Finally, you claim that your proposal is a foundational SpeechLM, but I cannot find the paper provides any explanation to state this point, which contradicts your response.
>
> While we did not explicitly use the term "foundational SpeechLM" in our paper, our method focuses on training speech connectors for SpeechLM and did not use instruction-tuning dataset. Moreover, our evaluation suite on spoken language understanding and text understanding should make it clear that our model is a foundational model rather than an instruction-fine-tuned one. Our comparison is also made with foundational SpeechLM like SpiritLM. We will add additional clarification in our paper to make this point clear. Thanks for raising this concern!
>
> We hope you could consider changing your rating criteria. Your points about paralinguistic/instruction-tuning are valid. However, there is no perfect work that can target all aspects of SpeechLM. We hope you could assess the quality of our work following our motivation and problem definition.

---

> > ### Comment · Reviewer_Bk9J · 2024-11-26
> >
> > Thanks to the author for further responding to my reply. I believe there may be some misunderstandings in your interpretation of my response, and it seems you may not have reviewed my reply very carefully. To help you better understand my response, I am providing additional clarifications here:
> >
> > > If you wish to hold us to a different standard by redefining/expanding the problem scope to paralinguistic information or instruction-finetuning stage, you must first discredit the importance of the problem we are solving (i.e., show that modality fusion or catastrophic forgetting is a trivial or already-solved problem for the scientific community).
> >
> > I never said that modality fusion and catastrophic forgetting are unimportant issues. At the very beginning of my response, I agreed that these two problems are very important. However, my point is that in the SSR-Connector paper, the focus is solely on the content understanding problem. This is akin to an end-to-end ASR + LLM model, and you cannot claim that this approach may address the key issues of Speech LMs (i.e., modality fusion and catastrophic forgetting).
> >
> > In addition, it only partially mitigates these issues at the content level. Addressing these tasks at the content level is not enough for Speech LMs (or, in my view, the benefit to Speech LMs is somewhat limited). Unless you can demonstrate that your approach can be easily adapted to other perspectives to help SpeechLM understand aspects beyond content, or that your model outperforms other end-to-end models on speech tasks (such as emotion), I don’t see how addressing these two issues from the perspective of content alone would be very meaningful.
> >
> > > Please refer to our evaluation on spoken language understanding and MMLU.
> > > The notion of “instruction fine-tuning phase is more important and practically valuable” is very problematic (a pre-trained LLM like Llama3 is not useful compared to llama3-instruct, does this mean pre-training method and details of Llama3 are not worth publishing if no instruction tuning result is reported?).
> >
> > You misunderstood my point. I didn't mean to say that foundation models are unimportant. What I meant is that you cannot prove your model is a good foundation model for Speech LM. Simply having strong content-understanding capabilities does not necessarily make it a good foundation model for speech. Since you claim your model is a good foundation model, please demonstrate that it can achieve better performance on certain speech-related tasks after some instruction-tuning, while also avoiding catastrophic forgetting issues regarding content understanding. Otherwise, I cannot agree that this is a good foundation model.
> >
> > Overall, I feel that the authors tend to overstate its contributions, including claims about being a "foundation model" and addressing the catastrophic forgetting issue in Speech LMs. While it does make some meaningful progress in mitigating the catastrophic forgetting problem at the content level, I am willing to raise the score to 5. However, I still lean toward rejecting this paper. Because mitigating the issue of catastrophic forgetting only at the content level offers limited novelty.

---

> > > ### Author Response · Authors · 2024-11-26
> > > **Response by Authors**
> > >
> > > Thank you for providing additional clarification about your previous response. We also thank you for raising the score to 5 to acknowledge that we made "some meaningful progress in mitigating the catastrophic forgetting problem at the content level".
> > >
> > > We believe we hold different opinions regarding the importance of problem A (modality fusion from a content/semantic perspective) and problem B (paralinguistic and other downstream tasks after instruction-tuning).
> > > - From our perspective, studying A itself is a sufficient contribution. This is because the previous effort in end-to-end SpeechLM suffers severely catastrophic forgetting (connector-based like Salmonn or unit-based like SpiritLM). We try to understand and alleviate this issue by using speech-text alignment.
> > > - From your perspective, B is essential and needs to be considered together with A when designing the system.
> > >
> > > We attempted to address your concern in B by adding in-context learning experiments on Expresso to demonstrate that paralinguistic information is still contained in the encoded representation, and we are happy to see that you're willing to increase the score to acknowledge our effort. Though we are still sorry to hear that you lean towards rejection, we believe at this point, it is more about subjective assessment of the importance of problems A and B and we understand that you hold different opinions from us. Nevertheless, we thank you for your time and constructive feedback that enhanced our work.

---

### Official Review · Reviewer_98Pj · 2024-11-02

**Soundness:** 2
**Presentation:** 2
**Contribution:** 2
**Rating:** 3
**Confidence:** 4

**Summary:**

This paper tackles the issue of catastrophic forgetting in the Language Model (LM) during speech fusion. The authors introduce the SSR-Connector method to enhance modality fusion. This approach utilizes a speech-text aligner and a decoder-only compressor to produce speech representations that align with the granularity of text embeddings.

**Strengths:**

The proposed method outperforms existing speech fusion mechanisms in terms of speech understanding.

**Weaknesses:**

The cost-performance ratio of the proposed method is lower when compared to a cascaded ASR+LLM system.
Please refer to the questions section regarding my queries.

**Questions:**

At the very beginning, it is mentioned that integrating speech often results in catastrophic forgetting of the primary pre-training. Have you quantified the severity of this issue? During the second stage of training where you fine-tune the Large Language Model (LLM), is there a risk of forgetting within the LLM itself? Specifically, I am referring to the LLM's performance (across various NLP tasks) before and after incorporating the speech modality.

Could you provide the specific figure for the original LaLLM3 as listed in Table 6?

Could you provide further explanation for the statement "lack the ability to support text or speech generation unless further instruction-finetuned" in the introduction section?

Given the option to use a decoder-only compressor and considering the overall model complexity, why not opt for a cascade system like Whisper+LLM? My assumption is that post the entire training process, the adapted representation z (as shown in Figure 3, right-hand side) will closely resemble the input embedding of the corresponding text (or sub-word units). This scenario might make the entire SSR-Connector resemble an Automatic Speech Recognition (ASR) system. In that case, why not simply use an ASR with text (or sub-word units) output directly?

Does the Speech Signal Representation (SSR) convey any audio/voice information beyond text? If so, have you conducted an analysis on this aspect? This is crucial as the community is progressing towards general audio Large Language Models as opposed to solely speech-based models. If not, similar to the previous question, what is the rationale behind incorporating an ASR system?

---

> ### Author Response · Authors · 2024-11-19
> **Response by Authors**
>
> We deeply appreciate the time and effort you have dedicated to reviewing this paper. We value your insights, and in response to your points of concern, we offer the following explanations:
>
> >Speech often results in catastrophic forgetting...Have you quantified the severity of this issue?
>
> Yes, to understand how modality fusion impacts the original LLM's performance on the text modality, we kindly refer you to our experimental results (MMLU performance) in Table 4, Table 6, and Figure 5. Here are some figures to provide a rough sense of catastrophic forgetting: For SpiritLM [1], the injection of speech tokens results in an MMLU drop from 65 to 53.5. In our method (see Section 5.1), vanilla fine-tuning decreases MMLU from 65 to 48.8 after 20k steps, whereas a multitask training setup maintains an MMLU of 63.1.
>
> Other prior works [2,3] also mention, though not always quantified, catastrophic forgetting in their studies. For instance, Moshi [2] observed a drop in MMLU from 54.3 to 49.8 after incorporating speech information.
>
> >is there a risk of forgetting within the LLM itself? Specifically, I am referring to the LLM's performance before and after incorporating the speech modality.
>
> After incorporating the speech modality, the LLM's performance on NLP tasks tends to degrade, as evidenced by the drop in MMLU. A key contribution of our work is to mitigate this degradation through our alignment-aware speech connector, which only results in a drop of 2 points compared to severe degradation from prior work like SpiritLM.
>
> However, if we have misunderstood your question, we kindly ask for further clarification on what you mean by "forgetting within the LLM itself."
>
>
> >Could you provide the specific figure for the original Llama3 as listed in Table 6?
>
> We kindly request more clarification on this question. Are you referring to the original (text-only) Llama3 model’s performance on MMLU in Table 6? If so, it is 65.3 (same as our stage1 distilled model because we did not modify Llama3 weight during stage 1 distillation)
>
>
> >Could you provide further explanation for the statement "lack the ability to support text or speech generation unless further instruction-finetuned"
>
> Thanks for raising this question. Here we are referring to the connector-based methods [3,4,5], which uses ASR format data to fine-tune the connector. After the connector is trained, the SpeechLM can only perform the specific kind of task it is trained on (e.g., speech recognition or translation) but cannot serve as a foundational SpeechLM like SpiritLM or our model. We also provide further comparison with connector-based model like SALMONN [3] in our "general response to all reviewers"
>
> >...In that case, why not simply use an ASR with text (or sub-word units) output directly?
>
> Thank you for the question. We agree that the cascaded ASR+LLM system is a straightforward and effective approach. However, from our perspective, modeling speech and text jointly within an end-to-end system is the future for SpeechLM. This approach allows the system to leverage information beyond text, which relates to your next question: *Does the Speech Signal Representation (SSR) convey any audio/voice information beyond text?*
>
> Currently, we still observe a significant performance gap between previous SpeechLM models and the cascaded baseline, as shown in Table 4. This observation motivates us to develop a more effective modality fusion approach to bridge the gap between an end-to-end SpeechLM and the cascaded system.
>
> >Does the Speech Signal Representation (SSR) convey any audio/voice information beyond text? If so, have you conducted an analysis on this aspect?
>
> Thank you for raising the concern regarding paralinguistic information. We have conducted additional experiments to demonstrate that our model retains information beyond text. We kindly refer you to the "general response to all reviewers" section for more details on these experiments. The results indicate that paralinguistic information is preserved in the representation, which opens up opportunities for future research to integrate our modality fusion method with other approaches to learn non-semantic information. Lastly, we would like to emphasize that the primary focus of our study is still on semantics. We believe that bridging the gap between the end-to-end SpeechLM and the cascaded system is meaningful in its own right.
>
>
> ---
>
> References:
>
> [1] Nguyen et al., (2024). Spirit LM: Interleaved Spoken and Written Language Model
>
> [2] Défossez et al., (2024). Moshi: a speech-text foundation model for real-time dialogue
>
> [3] Tang et al., (2024). Salmonn: Towards generic hearing abilities for large language models
>
> [4] Wu et al., (2023). On decoder-only architecture for speech-to-text and large language model integration
>
> [5] Yu et al., (2023). Connecting speech encoder and large language model for asr

---

> ### Author Response · Authors · 2024-11-28
> **Official Comment by Authors**
>
> Dear reviewer 98Pj,
>
> We sincerely appreciate your valuable suggestions for improving our paper and hope our response has effectively addressed your concerns. Should you have any additional suggestions or questions, please do not hesitate to let us know. We would greatly appreciate it if you could consider raising the rating. Thank you!
>
> Best regards,
> Authors

---

### Official Review · Reviewer_izXP · 2024-11-04

**Soundness:** 3
**Presentation:** 4
**Contribution:** 3
**Rating:** 8
**Confidence:** 4

**Summary:**

This paper proposes a new architecture with the corresponding loss design to connect speech embeddings to LLM. Thorough experiments are conducted to demonstrate the values of the proposed method.

**Strengths:**

The proposed method has some connections with the previous methods in speech downsampling and model distillation but is novel enough. The evaluation is very thorough.

**Weaknesses:**

Although the paper acknowledges the three categories of connector designs in Figure 1, it does not compare enough to Figure 1(a). Compared with the proposed 2-stage method, a single-stage or 2-stage baseline with a much simpler 1(a) design can motivate the more complex design in this paper.

Some experiment comparisons and explanations can be improved (see questions).

**Questions:**

1. need to explain why unitY2 based method is much better than others
2. "Even our weakest system (CHAR-CTC (Unit-based)) can outperform SPIRITLM ’s 10-shot result." the aligner in the proposed method is trained with paired ASR speech-text data, which is unfair to compare with SPIRITLM.
3. missing description of speech-MMLU

---

> ### Author Response · Authors · 2024-11-19
> **Response by Authors**
>
> We deeply appreciate the time and effort you have dedicated to reviewing this paper. We value your insights, and in response to your points of concern, we offer the following explanations:
>
> > …it does not compare enough to Figure 1(a).
>
> Thanks for raising the concern. For previous connector-based models (using compressor like Figure 1(a)), they either only focus on a special speech-to-text task like speech recognition or speech translation [1, 2], or necessitates instruction-tuning to support speech-text continuation [3]. If we compare with models like [1,2], their performance on spoken language understanding will be random as such models are only trained toward recognition/translation. For systems like SALMONN [3], to compare with their tasks, we need to perform instruction-tuning with our foundational SpeechLM, which is beyond the focus of this work.
>
> However, we added experiments to benchmark SALMONN on the Storycloze and Speech–MMLU dataset, and we kindly refer to our “general response to all reviewers” for details. We would like to highlight that our modality fusion mechanism is used to train speech-text foundation model, therefore our evaluation is focused on understanding and text-ability preservation rather than downstream applications, which is therefore more aligned with our compared prior works like SpiritLM, TWIST, VoxtLM etc.
>
>
> > A single-stage or 2-stage baseline with a much simpler 1(a) design can motivate the more complex design in this paper
>
> Thank you for the suggestion. We would like to clarify that the approach depicted in Figure 1(a) is not compatible with our two-stage training process because the compressed speech representation does not match the length of text tokens. Our ability to employ two-stage training stems from the fact that our compressed speech representation has the same shape as text tokens.
>
> Regarding the single-stage baseline, our preliminary experiments without distillation (as mentioned in lines 213-215) resulted in severe catastrophic forgetting, with MMLU dropping below 30 and suboptimal speech understanding. We will add further clarification on these points to enhance the clarity of the paper.
>
> > Need to explain why unitY2 based method is much better than others
>
> Thanks for the suggestion and we would like to refer you to section 4.3 for our aligner comparison. Basically unitY2 aligner is better because it produces better alignment (as verified by the word boundary error and word duration in Table 3) compared to CTC, which helps the stage 1 distillation to match speech to text representation.
>
> > The aligner in the proposed method is trained with paired ASR speech-text data, which is unfair to compare with SPIRITLM
>
> Thank you for your comment. We agree that leveraging segmentation from pre-trained aligners provides a strong prior for our SpeechLM to excel in the ASR task. However, we respectfully disagree with the notion that comparing our model with SpiritLM is unfair. Since both our SpeechLM and SpiritLM are foundational models, we believe it is appropriate to evaluate them on the same tasks. Note that even if SpiritLM is explicitly fine-tuned with ASR data (as shown in Table 5 of SpiritLM paper[4]), its 10-shot ASR performance (WER=6.0) is still worse than our best system.
>
>
> > Missing description of speech-MMLU
>
> Thank you for raising this concern. Currently due to page limitation, we provide a brief description of Speech-MMLU dataset from line 263-268, and put additional information in Appendix C. Please let us know if you need any further clarification/information about the Speech-MMLU dataset.
>
> ---
> References:
>
> [1] Wu et al., (2023). On decoder-only architecture for speech-to-text and large language model integration
>
> [2] Yu et al., (2023). Connecting speech encoder and large language model for asr.
>
> [3] Tang et al., (2024). Salmonn: Towards generic hearing abilities for large language models
>
> [4] Nguyen et al., (2024). Spirit LM: Interleaved Spoken and Written Language Model

---

> > ### Comment · Reviewer_izXP · 2024-11-20
> >
> > "A single-stage or 2-stage baseline with a much simpler 1(a) design" I refer to can we build a single-stage or 2-stage speech-text training baseline with much simpler 1(a) design but not with the proposed architecture

---

> > > ### Author Response · Authors · 2024-11-20
> > > **Response by Authors**
> > >
> > > Thanks for your clarification. Our proposed architecture can be seen as a special case of 1(a) where the compressor is alignment-aware. For a simpler 1(a) design, here are a few options:
> > >
> > > - Rule-based compressor [1] (using CNN to compress input). This approach is limited to ASR tasks after connector training.
> > > - Mean-pooling compressor: We experimented with an alignment-aware mean-pooling module to compress speech to match the length of text tokens. This method is not of good quality.
> > > - CIF-based compressor: since mean-pooling is not working, we tried the CIF-based compressor (and its performance is reported in the paper, see Table 2). It also does not work as well as our proposed SSR-connector.
> > >
> > > We hope our responses have addressed your question. If you have other special cases of 1(a) in mind, please feel free to give us more details for discussion.
> > >
> > >
> > > ---
> > >
> > > [1] Fathullah et al. (2023). Prompting Large Language Models with Speech Recognition Abilities

---

> > > > ### Comment · Reviewer_izXP · 2024-11-21
> > > >
> > > > have you done above with adding 2-stage training especially multitask finetune in the second stage? will that successfully keep the textual ability while enjoying simpler design in compressor?

---

> > > > > ### Author Response · Authors · 2024-11-21
> > > > > **Response by Authors**
> > > > >
> > > > > For CIF, as noted in Table 2, after stage-1 training, it has bad ASR word error rate and also bad performance on Speech-MMLU (5-shot accuracy of 43.7) so we did not pursue 2-stage training for it as it's first stage is already way worse than the other aligner.
> > > > >
> > > > > For simpler method like mean-pooling, unfortunately we did have the numbers for its performance on ASR or spoken language understanding anymore. This is the first system we tried, and we found it not performing well in first stage (which motivates the design of our Transformer-Decoder based pooling method).
> > > > >
> > > > > The takeaway is that: for CIF and mean-pooling, they are completely unusable with the single-stage training (directly trained with NLL). With 2-stage training, their performance lacks behind our proposed method by a large margin after distillation (stage 1), therefore we did not find it meaningful to pursue multitask fine-tuning (stage 2).
> > > > >
> > > > > We hope information above helps address your concerns and we will add clarification about these baselines in our paper to better motivate our design!

---

> ### Author Response · Authors · 2024-11-28
> **Official Comment by Authors**
>
> Dear Reviewer izXP,
>
> Thank you for acknowledging our responses and increasing the rating from 6 to 8! If you have any additional questions or suggestions, please do not hesitate to let us know.
>
> Best regards,
> Authors

---

### Author Response · Authors · 2024-11-19
**General Response to all reviewers**

We sincerely thank all the reviewers for their constructive feedback on our work. We are particularly pleased that several reviewers (izXP, Bk9J, j1u2) recognized the novelty of our modality fusion approach.


## Paralinguistic Information
We noted a common concern from reviewers 98Pj and Bk9J regarding the ability of our SpeechLM to leverage information beyond semantics. We would like to address this concern in our general response.
Firstly, we acknowledge the importance of utilizing paralinguistic information in end-to-end SpeechLM systems, and we agree that this is an area that requires further development. However, similar to most existing works on SpeechLM, our primary focus has been on semantic information. This is because **addressing catastrophic forgetting and improving speech understanding in end-to-end SpeechLM is a challenging and significant problem in its own right**. Our modality fusion approach has demonstrated superior performance compared to several strong baselines (such as SpirtLM, TWIST, VoxtLM, etc.) in both speech and text understanding tasks. Therefore, we believe it would be an overly stringent criterion to dismiss our work on the basis of not incorporating non-semantic information at this stage.

We still appreciate the reviewers for highlighting concerns about the non-semantic aspects of our work, which motivated us to conduct additional experiments. These experiments aimed to determine whether our SSR-based SpeechLM retains paralinguistic information, particularly in the context of in-context learning. Interestingly, we found that **our model, despite not being explicitly trained to use paralinguistic cues, can learn to differentiate between speech styles through in-context learning**.
To investigate this, we used the Expresso dataset [1], which features speeches in different styles (e.g., happy, sad, whispering, laughing). We conducted two sets of experiments:

- Whisper vs. Laugh: We tasked the model with identifying whether a speech was whispered or laughed. The prompt used was:
*"You are given speeches from two styles. Your task is to judge if the speech is a whisper or laugh. Here are some example speeches: [Speech]: {speech} [Style]: {whisper/laugh}..."*

- Happy vs. Sad: We asked the model to determine if the speech was delivered happily or sadly. The prompt was:
*"Listen to the following speech and judge if the speaker is happy or sad. Here are some examples: [Speech]: {speech} [Emotion]: {happy/sad}..."*

For each experiment, we provided our SpeechLM (trained with an SSR-connector based on the UnitY2 aligner) with 0, 1, 5, and 10 examples, and reported the accuracy (averaged over 10 runs) in the table below (We also benchmarked a cascaded system, Whisper + Llama3, for comparison):


|Task | Model  | 0-shot |1-shot| 5-shot | 10-shot |
|--------|---------|-------|-------|--------|---------|
|Whisper v.s. Laugh | Cascaded System | 51.6  | 52.1 | 52.2 | 54.7 |
| | Ours         | 49.6 | 62.4 |64.0 |75.9  |
| |
|Happy v.s. Sad | Cascaded System | 50  | 51.4 | 51.8 | 51 |
| | Ours         | 51.6 | 52.1 |52.2 | 54.7 |

With zero-shot prompting, the model generates random predictions because it hasn't been trained to use paralinguistic information. However, when given a few-shot examples, the model can learn from the prompts to make decisions, achieving surprisingly good results in distinguishing between whispering and laughing speech. In-context learning also helps to some extent in differentiating happy and sad speech, though not as effectively as distinguishing between whispering and laughing. We believe this is because the style difference between whispering and laughing is more pronounced. **The results indicate that paralinguistic information is still present in our model's representations and can be utilized**. Additionally, our approach complements previous methods that enable models to consider paralinguistic information, such as using expressive tokens from SpiritLM or emotion-relevant instruction tuning from SALMONN.

---

> ### Author Response · Authors · 2024-11-19
> **General Response to all reviewers (cont.)**
>
> ### Comparison with Connector-based Model (SALMONN)
> Besides concern over paralinguistic information, another common suggestion (reviewers izXP and Bk9J) was to compare our model with connector-based models like SALMONN [2]. We initially did not pursue this comparison because previous connector-based models are typically limited to specific speech-to-text tasks, such as speech recognition or translation [4,5], or they require instruction-tuning [2,3]. In contrast, our approach focuses on developing a foundational SpeechLM with high speech and text understanding, aligning more closely with prior works like SpiritLM, TWIST, and VoxtLM.
>
> However, to alleviate the reviewer's concern, we conducted experiments with SALMONN with our benchmarks: Storycloze (S->T) and Speech-MMLU, as SALMONN supports only the speech-to-text format. For Storycloze, we used the prompt: *<Speech><SpeechHere></Speech> Based on the audio, continue the story. <Text Continuation>*, and calculated the likelihood of good versus bad continuations for prediction.
>
> For Speech-MMLU, the prompt we used was: *<Speech><SpeechHere></Speech> Please listen to the audio and answer the question. Choose from the following options: {Choice String}.* Where the choice string covers four options from MMLU and we compute the model's likelihood for each option for prediction.
>
>
> The evaluation result is shown in the table below:
> | Model                   | Storycloze S->T | Speech MMLU (0-shot) |
> |-------------------------|-----------------|----------------------|
> | SALMONN                 | 63.3            | 25.3                 |
> | SpiritLM (Llama3)       | 61.6            | 40.5                 |
> | Ours (UnitY stage 1)    | 74.8            | 64.2                 |
> | Ours (Char-CTC stage 1) | 74.3            | 61.7                 |
> | Ours (Char-CTC stage 2) | 73.1            | 48.1                 |
>
>
> When comparing SALMONN with our method, we observe that SALMONN's performance is sub-optimal, particularly on the more challenging Speech-MMLU benchmark, where its performance is nearly random. It's important to note that SALMONN is fine-tuned on Vicuna-13B-v1.1 [6], which achieves an MMLU score of around 50. These results highlight the challenges of modality fusion and suggest that previous instruction-tuned SpeechLMs have experienced significant catastrophic forgetting.
>
> We hope these additional experiments and analyses could address reviewers’ concern and provide further justification for the significance of our work.
>
>
> ---
> References:
>
> [1] Nguyen et al., (2023). EXPRESSO: A Benchmark and Analysis of Discrete Expressive Speech Resynthesis
>
> [2] Tang et al., (2024). Salmonn: Towards generic hearing abilities for large language models
>
> [3] Chu et al. (2024). Qwen2-audio technical report
>
> [4] Wu et al., (2023). On decoder-only architecture for speech-to-text and large language model integration
>
> [5] Yu et al., (2023). Connecting speech encoder and large language model for asr
>
> [6] The Vicuna Team, (2023). Vicuna: An Open-Source Chatbot Impressing GPT-4 with 90%* ChatGPT Quality

---

### Meta-Review · Area_Chair_F7N4 · 2024-12-08

**Metareview:**

The paper introduces SSR-Connector, an approach designed to integrate speech data into pre-trained LLMs. It achieves this by aligning and compressing speech representations to match text embeddings, enhancing the fusion of speech and text modalities. SSR-Connector employs a two-stage training process: an initial distillation phase to align speech representations with text embeddings and fine-tuning to enhance cross-modal understanding. This approach demonstrates improvements in certain speech understanding benchmarks, surpassing prior frameworks like SpiritLM and VoxtLM.

The novelty of the paper is limited. The concepts of connector distillation and text-to-speech alignment have been explored in prior studies. Although these previous works differ from the current one, the authors have not compared their approach with these earlier methods to demonstrate its advantages. The training of the SSR connector is also similar to ASR, with the primary distinction being that LLM embeddings are used as the target. Consequently, reviewers have expressed concerns regarding the benefits of the proposed approach compared to an ASR+LLM cascade model. Furthermore, reviewers pointed out the absence of certain baseline comparisons.

**Additional Comments On Reviewer Discussion:**

Reviewer 98Pj, Reviewer Bk9J, and Reviewer j1u2 raised concerns that a straightforward pipeline of ASR followed by an LLM performs well, prompting questions about whether the proposed approach offers substantial advantages. Specifically, they noted that the SSR-Connector relies on alignment methods resembling ASR-like processing. The reviewers requested clearer distinctions from a cascaded baseline and a rationale for preferring an end-to-end SpeechLM over directly utilizing ASR transcripts. They also highlighted that the model's best performance depends on the UNITY2 model for alignment, which requires both speech and transcription inputs. Since transcripts are already available, they argued it might be feasible to train an ASR instead.

Author's reply: The authors contended that while cascaded systems serve as strong baselines, their approach represents progress toward a true end-to-end SpeechLM capable of leveraging non-textual cues.

Multiple reviewers (notably 98Pj, Bk9J) raised concerns that the proposed method focuses heavily on semantic content and may not leverage or preserve non-semantic (paralinguistic) features such as emotion, style, or speaker characteristics. They argued that a core advantage of end-to-end SpeechLMs over cascaded systems should be capturing richer acoustic cues beyond text. They questioned whether the SSR-Connector reduces to an ASR-like step and requested additional evidence that paralinguistic information is not lost.

Author's reply: In response to critiques on non-semantic aspects, the authors conducted new few-shot experiments using the Expresso dataset to test whether their model can distinguish different speech styles (e.g., whisper vs. laugh, happy vs. sad). Although zero-shot performance was random, the model improved significantly with a few examples, suggesting that some paralinguistic cues are retained and can be utilized via in-context learning.

Reviewer Bk9J suggested comparing more models, such as SALMONN and Qwen-Audio.

Author's reply: They conducted additional evaluations on SALMONN and found it performed poorly on challenging tasks like Speech-MMLU.

Note from AC: However, they did not compare their model with Qwen2-Audio, which has demonstrated better performance than SALMONN (as shown in https://arxiv.org/abs/2409.20007, Table 1).

Reviewer j1u2 pointed out that LLaMa2 was used for the cascade system, while LLaMa3 was used in the proposed method, which seems unfair.

Author's reply: They believe that LLaMa2 and LLaMa3 would not differ significantly in terms of text continuation.

Note from AC: Their evaluation also includes MMLU, which LLaMa2 and LLaMa3 may have different performance.


The authors clarified that their goal is to create a strong "foundational" SpeechLM that can later be instruction-tuned or extended to handle paralinguistic tasks. However, Reviewer Bk9J noted that no experiments demonstrate whether the proposed model can serve as an effective foundation model.

AC's opinion:
The reasons the paper is not ready for publication are outlined below:
+ Similarity to previous work: Some similar ideas have already been proposed. For example, the concept of connector distillation has been explored in prior works (e.g., https://arxiv.org/abs/2406.05968, https://arxiv.org/abs/2405.19041), and text-to-speech alignment has also been investigated (e.g., https://arxiv.org/abs/2307.03917). While the proposed approach is not exactly the same, the authors do not provide comparisons with these previous methods.
+ Training method concerns: The training of the SSR connector is similar to ASR, with the primary difference being the use of LLM embeddings as the target. Consequently, it is natural for reviewers to question the benefits of this proposed approach over an ASR+LLM cascade model. The authors address this issue from two aspects: (1) The model can serve as a foundation model for subsequent fine-tuning. However, the paper lacks related experiments to support this claim. (2) The authors suggest that the proposed model can capture paralinguistic information to some extent through in-context learning. Results indicate that paralinguistic information remains present in the model's representations and can be utilized. This additional experiment partially addresses the concern. Including more analysis in this direction in future revisions of the paper is recommended.
+ Missing required baselines: Several models already aim to align speech with text LLM inputs. The original paper does not compare its approach with these models. During the rebuttal, the authors compared their method with SALMONN, but given the performance, Qwen2-Audio could serve as a more suitable baseline.

---

### Decision · Program_Chairs · 2025-01-22

Reject